# Comprehensive Investigation of Stereoselective Food Drug Interaction Potential of Resveratrol on Nine P450 and Six UGT Isoforms in Human Liver Microsomes

**DOI:** 10.3390/pharmaceutics13091419

**Published:** 2021-09-07

**Authors:** Seung-Bae Ji, So-Young Park, Subin Bae, Hyung-Ju Seo, Sin-Eun Kim, Gyung-Min Lee, Zhexue Wu, Kwang-Hyeon Liu

**Affiliations:** 1BK21 FOUR KNU Community-Based Intelligent Novel Drug Discovery Education Unit, Daegu 41566, Korea; wltmdqo2377@naver.com (S.-B.J.); soyoung561@hanmail.net (S.-Y.P.); bsb960908@naver.com (S.B.); hlhl103@naver.com (H.-J.S.); hjkopsty@gmail.com (S.-E.K.); lgm00179@naver.com (G.-M.L.); 2Research Institute of Pharmaceutical Sciences, College of Pharmacy, Daegu 41566, Korea; 3Mass Spectrometry Based Convergence Research Institute, Kyungpook National University, Daegu 41566, Korea

**Keywords:** cytochrome P450, food drug interactions, resveratrol, stereoselectivity, uridine 5′-diphosphoglucuronosyl transferase

## Abstract

The stereoselectivity of the food drug inhibition potential of resveratrol on cytochrome P450s and uridine 5′-diphosphoglucuronosyl transferases was investigated in human liver microsomes. Resveratrol enantiomers showed stereoselective inhibition of CYP2C9, CYP3A, and UGT1A1. The inhibitions of CYP1A2, CYP2B6, and CYP2C19 by resveratrol were stereo-nonselective. The estimated *K*_i_ values determined for CYP1A2 were 13.8 and 9.2 μM for *trans*- and *cis*-resveratrol, respectively. *Trans*-resveratrol noncompetitively inhibited CYP3A and UGT1A1 activities with *K*_i_ values of 23.8 and 27.4 μM, respectively. *Trans*-resveratrol inhibited CYP1A2, CYP2C19, CYP2E1, and CYP3A in a time-dependent manner with *K*_i_ shift values >2.0, while *cis*-resveratrol time-dependently inhibited CYP2C19 and CYP2E1. The time-dependent inhibition of *trans*-resveratrol against CYP3A4, CYP2E1, CYP2C19, and CYP1A2 was elucidated using glutathione as a trapping reagent. This information helped the prediction of food drug interaction potentials between resveratrol and co-administered drugs which are mainly metabolized by UGT1A1, CYP1A2, CYP2C19, CYP2E1, and CYP3A.

## 1. Introduction

Resveratrol (3,5,4′-Trihydroxystilbene, RVT), a polyphenolic phytoalexin, is a common constituent found in many plants or beverages such as berries, grapes, peanuts, soy, and in wine, cranberry, and grape juices, as well as in many other plant products [1,2]. It is considered to be one of the anti-inflammatory and anti-oxidant constituents in grape juice and red wine [3]. Owing to various pharmacological effects of RVT [4,5,6], this molecule has been consumed in the forms of dietary supplements.

As the consumption of dietary supplements for health benefits has been increasing every year, food drug interactions (FDI) have been a focus of intensive research. In 2019, the global dietary supplement market was valued at 163 billion [7]. Most of these FDI might be attributed to the induction or inhibition of drug transporters and drug-metabolizing enzymes [8]. Grapefruit juice is a well-known example of such FDI, and bergamottin is considered to be the major inhibitor of CYP3A [9]. *Schisandra* extracts also markedly increased the plasma concentration of tacrolimus by inhibiting the CYP3A4 isoform in liver transplant patients [10].

Inhibition of cytochrome P450 enzymes (P450s) by *trans*-resveratrol (tRVT) have been extensively studied, including the inhibition of CYP3A4, CYP2C19, CYP2C9, and CYP1A2 [11,12,13]. tRVT has also been reported to irreversibly inhibit the activities of CYP1A2 [14], CYP2C19 [15], CYP2E1 [16], and CYP3A4 [17] in human liver microsomes (HLMs). The inhibition of P450 activity by RVT could lead to safety issues by altering the clearance of co-administered drugs [13]. In clinical trials, tRVT has been shown to affect the pharmacokinetics of carbamazepine, a CYP3A substrate [18], and chlorzoxazone, a CYP2E1 substrate [19], due to its irreversible inhibition of CYP2E1 and CYP3A. Despite extensive studies on inhibitory potential of tRVT against P450s, there is only one study on uridine 5′-diphosphoglucuronosyl transferase enzymes (UGTs). tRVT inhibited UGT1A1-mediated 7-ethyl-10-hydroxycamptothecin (SN-38) glucuronidation with a *K*i value of 6.2 μM in human recombinant UGT1A1 supersome [20].

Resveratrol naturally occurs in two isomeric forms, *trans*- and *cis*-resveratrol (cRVT) (Figure 1). RVT can isomerize from the *trans*- to the *cis*-isomer under ultraviolet-light exposure [21], therefore, cRVT is also mainly present in red wine and grape juice together with tRVT. tRVT concentrations in red wine ranged from 0.22 to 47.8 μM (0.05~10.9 μg/mL), whereas cRVT ranged from 0.18 to 52.6 μM (0.04~12.0 μg/mL) [22,23,24]. Despite the wide observation of cRVT in commercial wine and grape juice products, studies on the inhibitory potential of cRVT against drug-metabolizing enzymes have been limited. cRVT inhibited CYP2C19-mediated *S*-mephenytoin 6-hydroxylation and CYP3A-mediated testosterone 6β-hydroxylation with IC_50_ values of 41.1 and 4.5 μM, respectively, whereas it had negligible effects on other P450s (IC_50_ > 100 μM) [1]. Though the time-dependent P450 inhibitory effects of tRVT are well known, no reports have been reported on cRVT. In addition, no detailed studies have been performed to elucidate the inhibition mode and potency of P450s and UGTs by cRVT.

A chiral phytochemical can stereoselectively modulate P450-catalyzed biotransformation. For example, *R*-naringenin has a 2-fold more potent inhibition for CYP2C9 and CYP3A inhibition compared with that of the *S*-enantiomer [25]; (-)-tetrahydropalmatine significantly inhibits CYP2D6-mediated dextromethorphan *O*-demethylase activity 15 times more extensively than (+)-tetrahydropalmatine [26], and quinidine stereoselectively inhibits CYP2D6-mediated debrisoquine 4-hydroxylation as much as quinine [27,28]. Until now, little information is available on the stereoselective inhibition of RVT on the UGT and P450 activities.

The four objectives of this study were: (1) to investigate the inhibition mode and kinetics of cRVT against six UGTs and nine P450s in HLMs; (2) to elucidate the time-dependent inhibition (TDI) of cRVT against P450s; (3) to compare the inhibitory potential of cRVT and tRVT against these enzymes, and (4) to elucidate the TDI mechanism of RVT in human recombinant P450 isoforms (rP450s).

## 2. Materials and Methods

### 2.1. Chemicals and Reagents

*Trans*-resveratrol (tRVT, purity ≥ 99%), *cis*-resveratrol (cRVT, purity ≥ 98%), dehydronifedipine, diclofenac, 6-hydroxychlorzoxazone, 4-hydroxydiclofenac, 4-hydroxymephenytoin, *S*-mephenytoin, midazolam, mycophenolic acid (MPA), nifedipine, CDCA-24-acyl-β-glucuronide, 7-ethyl-10-hydroxy camptothecin (SN-38) glucuronide, MPA-β-d-glucuronide, and *N*-acetylserotonine (*N*-ASER)-β-d-glucuronide were obtained from Toronto Research Chemicals (Toronto, ON, Canada). Acetaminophen, amodiaquine, bupropion, chenodeoxycholic acid (CDCA), chlorzoxazone, dextromethorphan, dextrorphan, 6-hydroxybupropion, 7-hydroxycoumarin, *N*-ASER, *N*-desethylamodiaquine, naloxone, nifedipine, phenacetin, trifluoperazine (TFP), estrone-β-d-glucuronide, naloxone-β-d-glucuronide, and TFP-β-d-glucuronide were obtained from Sigma-Aldrich (St. Louis, MO, USA). SN-38 and 1′-hydroxymidazolam were provided by Santa Cruz Biotechnology (Dallas, TX, USA) and Cayman Chemicals (Ann Arbor, MI, USA), respectively. Pooled HLMs (XTreme 200) were purchased by XenoTech (Lenexa, KS, USA). rP450 (rCYP3A4, rCYP2E1, rCYP2C19, and rCYP1A2) were obtained from SPMED (Busan, Korea).

### 2.2. Stereoselective Inhibition of Resveratrol against Cytochrome P450 Activity

The stereoselective inhibition of RVT on the activity of nine cytochrome P450 (P450) isoforms were evaluated as previously described with minor modifications [29,30]. RVT and each P450 substrates were dissolved in methanol. The microsomal incubation was performed in 1.5 mL amber tubes using two substrate cocktail sets (set A: amodiaquine, bupropion, dextromethorphan, diclofenac, *S*-mephenytoin, and phenacetin; set B: chlorzoxazone, coumarin, midazolam, and nifedipine) (Table 1). These substrate cocktails were validated by comparison of the inhibition obtained from incubation of each individual index substrate alone and the substrate cocktail in our previous study [29,30]. The incubation mixture included RVT (0, 0.5, 2, 5, 20, and 50 µM), P450 probe substrate cocktail set, 0.1 M potassium phosphate buffer, and 0.25 mg/mL HLMs. After preincubation (5 min, 37 °C), the reaction initiated upon the addition of an NADPH-generating system (1.3 mM β-NADP^+^, 3.3 mM MgCl_2_, 3.3 mM G6P, and 1.0 unit/mL G6PDH). Following further incubation for 10 min, acetonitrile containing internal standard (IS, 7 nM trimipramine) was added to terminate the reaction. After centrifugation (4 °C, 5 min, 14,000 rpm), the aliquots of supernatants were assayed by liquid chromatography-tandem mass spectrometry (LC-MS/MS). The experiment of RVT was conducted in amber tubes protected from light in order to avoid isomerization. All microsomal incubations were performed in triplicate.

The TDI of RVT against nine P450s was examined using an IC_50_ shift assay [31]. Each RVT (0~50 μM) was pre-incubated with HLMs for 30 min at 37 °C in the presence of an NADPH-generating system. After preincubation, P450 probe substrate cocktail set was added to start the reaction prior further incubation for 10 min. Afterwards, it was carried out the same as the above experimental method.

### 2.3. Stereoselective Inhibition of Resveratrol against Uridine 5′-Diphosphoglucuronosyl Transferase Activity

The stereoselective inhibition of RVT on the activity of six UGT isoforms were examined as previously described with minor modifications [32,33]. The incubation mixture included 25 µg/mL alamethicin, RVT (0, 0.5, 2, 5, 20, and 50 µM), UGT probe substrate cocktail set, 0.1 M tris-HCl buffer, and 0.25 mg/mL HLMs. After preincubation (5 min, 37 °C), the reaction started after the addition of 5 mM UDPGA. Following further incubation for 1 h, ice-cold acetonitrile containing IS (250 nM estrone-β-d-glucuronide) was added to terminate the reaction. After centrifugation (4 °C, 5 min, 14,000 rpm), the supernatants were filtered and assayed by LC-MS/MS (Table 1).

### 2.4. Kinetic Characterization of Resveratrol on CYP3A, CYP2E1, CYP2C19, and CYP1A2 in HLMs

We used HLMs to estimate the constants and mechanisms for tRVT and cRVT inhibition of CYP1A2 and CYP3A which had IC_50_ values less than 20 µM. Each RVT (0~50 μM) was added into the reaction solutions, each of which contained concentrations of midazolam (0.1, 0.5, and 2 μM), and nifedipine (0.2, 1, and 5 µM), and phenacetin (5, 20, and 50 μM). The other conditions were similar to those reported for the P450 inhibition study. The inhibition constants for RVT inhibition of CYP1A2, CYP2C19, CYP2E1, and CYP3A were also evaluated after 30 min preincubation with RVT in HLMs. Each RVT (0~50 μM) was added into the reaction solutions, each of which contained concentrations of midazolam (0.1, 0.5, and 2 μM), and nifedipine (0.2, 1, and 5 µM), phenacetin (5, 20, and 50 μM), *S*-mephenytoin (20, 40, and 100 µM), and chlorzoxazone (1, 5, and 20 μM). The concentrations of acetaminophen, 4-hydroxymephenytoin, 6-hydroxychlorzoxazone, 1′-hydroxymidazolam, and dehydronifedipine were assayed by LC-MS/MS as previously described [29,30]. The lower limits of quantification for acetaminophen, 4-hydroxymephenytoin, 6-hydroxychlorzoxazone, 1′-hydroxymidazolam, and dehydronifedipine were 1, 5, 5, 10, and 0.5 nM, respectively. The inter-assay precision values for all of the samples were less than 12.6%.

### 2.5. Kinetic Characterization of trans-Resveratrol on UGT1A1 in HLMs

We used HLMs to elucidate the constants and mechanisms for tRVT inhibition of UGT1A1 which had IC_50_ values less than 20 µM. tRVT (0~50 μM) was added into the reaction solutions, each of which contained concentrations of SN-38 (0.5, 2, and 10 μM). The concentrations of SN-38 glucuronide were measured by LC-MS/MS as previously described [32]. The lower limits of quantification for SN-38 glucuronide was 5 nM, respectively. The inter-assay precision values for all of the samples were less than 10.7%.

### 2.6. Characterization of Glutathione Conjugates of Resveratrol in Recombinant Cytochrome P450 Isoforms

tRVT (20 μM) was incubated (37 °C, 45 min) with rP450s (20 pmol/mL) in 100 mM phosphate buffer in the presence of glutathione (10 mM) and NADPH (1 mM). Control incubations in the absence of glutathione and NADPH were performed. Incubations were stopped by the addition of acetonitrile. The supernatants were concentrated and reconstituted with methanol (100 μL). Samples were assayed by LC-high resolution mass spectrometry (LC-HRMS) [34].

### 2.7. LC-MS/MS Analysis

All samples were assayed using a Shimadzu LC-MS 8060 tandem mass spectrometer combined with a Nexera X2 liquid chromatography equipped with an electrospray ionization device (Shimadzu, Kyoto, Japan). Nine P450- and six UGT-isoform specific metabolites were separated on a Kinetex XB-C18 column (100 Å, 2.6 μm, 100 × 2.1 mm; Phenomenex, Torrance, CA, USA). The mobile phase consisted of 0.1% formic acid (FA) containing water (A) and 0.1% FA containing acetonitrile (B), and elution condition was set as follows: the gradient was maintained with 8% B for 0.5 min and then linearly increased to 60% in 5 min and held for 1 min, after linearly reduced to 8% in 6.1 min; finally, the original gradient was applied and maintained in 9 min [30]. Gradient elution conditions were set as follows: 0% B, 30% B (0–1 min), 50% B (1–5 min), and 0% B (5.1–8 min) [32]. The flow rate was 0.2 mL/min. Ionization voltages in positive and negative ionization modes were 4000 V and −3500 V, respectively. Quantitation was conducted in selected reaction monitoring (SRM) modes for each metabolite (Table 1). 

Using a Vanquish HPLC combined with a QExactive Focus Orbitrap mass spectrometer (Thermo Fisher Scientific Inc., Waltham, MA, USA), we determined the glutathione adducts produced by rP450s. A Kinetex C18 column was also used to separate the analytes. The mobile phase was set as 80% acetonitrile in water containing 0.1% FA. The flow rate was set at 0.2 mL/min. Data acquisition was carried in the total ion scan mode (*m/z* 100–700) with a resolution of 70,000, and MS/MS spectra were acquired in the product ion scan mode (*m/z* 50–580) at a resolution of 17,500. Parallel reaction monitoring (PRM) conversion *m/z* 550.1495 was used for the identification of GSH conjugate.

### 2.8. Data Analysis

All results were acquired from three replicates in different microsomal incubations. IC_50_ values were determined by nonlinear regression analysis using WinNonlin (Pharsight, Mountain View, CA, USA). We used WinNonlin to estimate the apparent kinetic parameters of inhibitory activity (*K*_i_) and the type of inhibitory activity by several criteria, including visual inspection of Lineweaver–Burk double reciprocal plots, Dixon plots, and secondary plots of Lineweaver–Burk plots versus RVT concentrations in each inhibitory model [35,36].

## 3. Results and Discussion

### 3.1. Stereoselective Inhibition of Cytochrome P450 Isoform Activities by Resveratrol

To elucidate the stereoselective inhibition of RVT, we determined the inhibitory potential of tRVT and cRVT against P450 isoform activities using HLMs (Table 2). tRVT and cRVT inhibited CYP1A2-mediated phenacetin *O*-deethylation (12.9 vs. 14.0 μM), CYP2B6-mediated bupropion hydroxylation (34.8 vs. 28.1 μM), and CYP2C19-mediated *S*-mephenytoin hydroxylation (48.4 vs. 46.5 μM) in a stereoselectivity independent manner, whereas they stereoselectively inhibited CYP2C9-mediated diclofenac hydroxylation (36.2 vs. >50.0 μM), CYP3A-mediated midazolam hydroxylation (29.2 vs. >50.0 μM), and CYP3A-mediated nifedipine dehydrogenation (19.6 vs. >50.0 μM). The inhibitory effect of tRVT on CYP1A2 was consistent with the results of a previous study (IC_50_ = 7.8 μM) [1]. The inhibitory potential of tRVT on CYP2B6 (IC_50_ = 34.8 μM) and CYP2C9 (IC_50_ = 36.2 μM) were slightly lower than previous results (IC_50_ > 50.0 μM) [11,12]. This discrepancy could be due to differences in reaction conditions in the enzyme source (HLM vs. rCYP2B6 or rCYP2C9) or the CYP2B6 probe substrates (bupropion vs. 7-benzyloxyresorufin). tRVT showed a nearly 2-fold stereoselectivity for CYP3A inhibition compared to cRVT. Similar stereoselective inhibitory effects of RVT on CYP3A were also demonstrated when the inhibitory effects of tRVT and cRVT against CYP3A-mediated testosterone hydroxylase activity was evaluated to have 1.7-fold selectivity [1]. The inhibition of CYP2A6, CYP2C8, CYP2D6, and CYP2E1 activities by *trans*- and cRVT was negligible (IC_50_ > 50.0 μM). Hyrsova et al. (2019) also reported that RVT had negligible inhibitory effects on CYP2A6, CYP2D6, and CYP2E1 activities [1]. 

Several phytochemicals including naringenin [25], tetrahydropalmatine [26], and quinidine [27] have shown to be time-dependent inhibitors of P450 enzymes. We examined the effect of incubation time on the inhibitory potential (IC_50_ values) of RVT on nine P450s (Table 2). A test compound with an IC_50_ fold-shift decrease ≥ 1.5 is considered to be a time-dependent inhibitor as suggested by Seo et al. [37] and Awortwe et al. [38]. Previous studies have elucidated that tRVT inhibits CYP1A2 [39], CYP2C19 [15], CYP2E1 [16], and CYP3A4 [15] activities in a NADPH- and time-dependent manner when co-incubated with rP450s or HLMs. Our data confirm previous findings and extend the time-dependent inhibitory ability of cRVT on P450s. tRVT inhibited CYP1A2, CYP2C19, CYP2D6, CYP2E1, and CYP3A in a time-dependent manner with IC_50_ shift values >1.5 (Table 2). The IC_50_ values of tRVT for CYP1A2, CYP2C19, and CYP2E1 activities were 10.8-, 6.0-, and 5.0-fold higher for reversible inhibition than for TDI, respectively. Therefore, CYP1A2 was the most susceptible to TDI by tRVT among these P450s. cRVT also time-dependently inhibited CYP1A2, CYP2C19, and CYP2E1, and it showed time-independent inhibition on CYP3A, CYP2D6, CYP2C8, CYP2B6, and CYP2A6 (Table 2). In HLMs, the rank order of the inhibitory potentials of cRVT as evaluated by IC_50_ values was CYP1A2 > CYP2E1 > CYP2C19 > CYP2B6 > CYP2A6, CYP2C8, CYP2C9, CYP2D6, and CYP3A. This indicated that CYP1A2, CYP2C19, and CYP2E1 were the most sensitive to inhibition by cRVT. In the TDI study, tRVT and cRVT stereoselectively inhibited CYP1A2-mediated phenacetin deethylation (1.21 vs. 7.20 μM), CYP2C19-mediated *S*-mephenytoin hydroxylation (8.13 vs. 19.8 μM), CYP3A-mediated midazolam 1′-hydroxylation (16.6 vs. >50.0 μM), and CYP3A-mediated nifedipine dehydrogenation (5.62 vs. >50.0 μM) with IC_50_ differences of >2.4-fold (Table 2). 

### 3.2. Stereoselective Inhibition of Uridine 5′-Diphosphoglucuronosyl Transferase Isoform Activities by Resveratrol

UGT inhibition by phytochemicals is known to be one of the most important factors for FDI. For example, pretreatment with UGT1A1 inhibitor psoralidin, a natural phenolic component found in the seeds of *Psoralea corylifolia*, increased the toxicity of irinotecan, an UGT1A1 substrate, as indicated by the severe colon histology damage in mice [40]. The plasma concentrations of emodin were significantly increased by pretreatment with stilbene glucoside, indicating that stilbene glucoside significantly affected the pharmacokinetics of emodin through the inhibition of UGT1A8 mRNA expression [41]. Pretreatment with soybean induces the induction of UGT, resulting in a decrease of the bioavailability of valproic acid [42]. However, there is still very limited data on the UGT-mediated drug interaction potential of resveratrol. Therefore, we investigated the stereoselective inhibitory potential of RVT for six UGTs’ isoform activities using HLMs (Table 3). tRVT inhibited UGT1A1-mediated SN-38 glucuronidation with an IC_50_ value of 9.57 μM, similarly to a previous finding (*K*_i_ = 6.2 μM) [20] in a stereoselective manner. The inhibition of the other five UGTs by tRVT and cRVT was considered negligible (IC_50_ > 50.0 μM). 

### 3.3. Kinetic Characterization of trans-Resveratrol and cis-Resveratrol Inhibition against CYP1A2, CYP2C19, CYP2E1, CYP3A, and UGT1A1 in HLMs

RVT inhibited microsomal CYP1A2, CYP2C19, CYP2E1, CYP3A, and UGT1A1 activities with IC_50_ values below 10 µM, therefore, we sought to clarify its inhibitory mechanism (Table 4). tRVT and cRVT competitively inhibited CYP1A2 activities with *K*_i_ values of 13.8 (r^2^ = 0.995) and 9.19 µM (r^2^ = 0.995), respectively. tRVT noncompetitively inhibited CYP3A and UGT1A1 activities with *K*_i_ values of 23.8 (r^2^ = 0.998) and 27.4 µM (r^2^ = 0.995), respectively, while cRVT did not inhibit any of these two enzymes (Figure 2 and Table 4). We also examined the effect of incubation time on the inhibitory potential (*K*_i_ values) of RVT on four P450s (Table 4). tRVT inhibited CYP1A2, CYP2C19, CYP2E1, and CYP3A in a time-dependent manner with *K*_i_ values of 1.62, 9.78, 9.11, and 1.02 μM, respectively (Figure 3 and Appendix A), while cRVT time-dependently inhibited CYP1A2, CYP2C19, and CYP2E1 with *K*_i_ values of 9.46, 21.8, and 8.08 μM, respectively (Figure 4 and Appendix A). The inhibition constant of tRVT on CYP1A2 was similar to Fairman et al.’s findings (*K*_i_ = 2.2~3.3 μM) [39]. The inhibition constants of tRVT on CYP2C19 (*K*_i_ = 9.78 μM) and CYP2E1 (*K*_i_ = 9.11 μM) were higher than previous results (*K*_i_ = 3.3 and 2.1 μM, respectively) [15,16], while that of tRVT on CYP3A (*K*_i_ = 1.0~4.6 μM) was lower than previous data (*K*_i_ = 20 μM) [43]. This discrepancy might be due to differences in reaction conditions in the enzyme source (HLM vs. rCYP2C19), the CYP2E1 probe substrates (chlorzoxazone vs. p-nitrophenol) or the CYP3A index substrates (nifedipine or midazolam vs. testosterone). In previous results, tRVT was able to inhibit CYP1A2-mediated tacrine hydroxylation [39] and CYP1A2-mediated ethoxyresorufin *O*-demethylation [14] activity with *K*_i_ values of 2.2 and 8.5 μM, respectively, in a substrate-dependent manner in HLMs. The inhibition constants of tRVT against CYP1A2, CYP2C19, CYP2E1, and CYP3A were 5.0-fold higher for reversible inhibition than for TDI. Therefore, TDI was considered to be the main reason responsible for CYP3A, CYP2E1, CYP2C19, and CYP1A2 inhibition by tRVT (Table 4). Unlike tRVT, cRVT did not show TDI against CYP1A2 and CYP3A.

### 3.4. Characterization of Glutathione Conjugates of trans-Resveratrol in Recombinant Cytochrome P450 Isoforms

RVT is known to be converted to reactive quinone methides by CYP3A4 [43,44] or CYP1A2 [34,45]. These quinone methide intermediates might react with CYP3A4 or CYP1A2 through covalent modification with these P450s [34,43]. The formation of a reactive intermediate-P450 complex has been reported to play an essential role in the TDI of P450 by RVT. Glutathione might be used as a trapping agent to identify quinone methide because the latter is unstable and cannot be directly identified [34]. The glutathione conjugates with the oxidized intermediate of RVT were identified in human liver microsomal incubation samples [34]. In this study, tRVT showed TDI of CYP1A2, CYP2C19, CYP2E1, and CYP3A activities with a *K*_i_ shifting > 5.0 in HLMs (Table 4). 

To elucidate the TDI mechanism of tRVT against CYP1A2, CYP2C19, CYP2E1, and CYP3A, tRVT was incubated with rP450s in the presence of NADPH and glutathione. LC-HRMS analyses indicated that there was one glutathione conjugate ([M+H]^−^, *m/z* 550.1495, t_R_ = 0.89 min) formed in rCYP3A4. UPLC-HRMS analyses of the peak responsible for this GSH conjugate displayed a protonated molecule [M+H]^−^ at *m/z* 550.1495 (mass error < 2 ppm), 323 Da higher than that of tRVT. This suggested that tRVT first oxidizes before it is being conjugated with one molecule of glutathione (MW = 307.3). The product ion scan spectrum of the glutathione conjugate by fragmenting *m/z* 550.1495 through collision generated characteristic daughter ions at *m/z* 428.1128 suggests the loss of a methyl dihydroxybenzene residue (−122 Da) (Figure 5). The fragment ion of *m/z* 306.0762 and 272.0885 was produced by glutathione moiety and cleavage of the cysteinyl C-S bond, respectively. The fragment ions observed from the glutathione moiety and cleavage of the cysteinyl C–S bond are the most typical ions found in glutathione conjugates [46,47]. The glutathione conjugate was also observed in the incubation samples with rCYP1A2, rCYP2C19, and rCYP2E1 (Figure 5). Our results showed that CYP1A2, CYP2C19, CYP2E1, and CYP3A4 were involved in the formation of reactive quinone methide of tRVT (Figure 5).

### 3.5. Evaluation of Food Drug Interaction Potential of Resveratrol

In previous studies, red wine and tRVT were found to change the pharmacokinetics of drugs which are substrates of CYP3A [48]. Zhan et al. (2015) reported that the multiple dose of RVT (100 or 200 mg/kg) significantly increased the AUC and *C*_max_ of aripiprazole (oral, 3 mg/kg) in rats [49]. Animal studies also showed that a seven-day co-administration with RVT (100 mg/kg) significantly increased oral pharmacokinetics of alogliptin, saxagliptin, and sitagliptin in rats through CYP3A inhibition [50]. In addition, four weeks of RVT dosing (1 g, once daily) inhibited the phenotypic indices of CYP2C9, CYP2D6, and CYP3A, and increased the metabolic ratio (caffeine/paraxanthine ratio) of CYP1A2 in healthy subjects [51].

In contrast to the extensive studies on FDI with CYP3A substrates, there are limited data on FDI with other P450s. We anticipated the clinical FDI risk induced by RVT based on each of the IC_50_ values. tRVT inhibited CYP2C19 and CYP2E1 activities with IC_50_ values of 8.1 and 9.8 μM, similar to CYP3A inhibition (IC_50_ = 5.6~16.6 μM) in a concentration- and time-dependent manner. Considering that tRVT participates in the pharmacokinetic intervention of dipeptidyl peptidase-4 inhibitors [50] and diltiazem [52] by inhibiting CYP3A-mediated biotransformation in rats, tRVT might interact with CYP2E1 or CYP2C19 substrate drugs such as clopidogrel [53], omeprazole [54], acetaminophen [55], and theophylline [56]. cRVT may also interact with CYP2E1 or CYP2C19 substrate drugs, because their inhibitory potential (IC_50_ = 10.8~19.8 μM) is stronger than CYP2C9 inhibition (IC_50_ = 23.8 μM). A dose of RVT received from food supplements may reach up to 2–5g/day, and RVT concentrations in plasma, after oral administration, may achieve micromolar concentrations [13,57]. In rats, oral administration of tRVT (100 mg/kg) resulted in a significant increase of *S*-warfarin plasma concentration and international normalized ratio through CYP2C9 inhibition [58]. In addition, we again predicted in vivo FDI potential of RVT using the volume per dose index (VDI (L) = RDI/IC_50_, RDI means recommended daily intake), volume in which the daily dose would be dissolved to reach the corresponding IC_50_ concentration [59]. The VDI cut-off value for enzymes present in the liver is 5.0 L [60], and the recommended daily intake value for RVT is 450 mg per day [61]. VDI values of tRVT for inhibiting CYP2C19 and CYP2E1 were 243 and 202 L, respectively, while those of cRVT were 183 and 100 L, respectively. The calculated VDI values of RVT exceeded 100 L per unit dose with *S*-mephenytoin and chlorzoxazone, indicating RVT might have potential to inhibit biotransformation by CYP2C19 and CYP2E1 enzyme in vivo. However, clinical studies are needed to assess whether RVT affects drug metabolism by CYP2C19 and CYP2E1 enzymes in vivo. 

An in vivo FDI via the inhibition of a drug-metabolizing enzyme might occur if the ratio of the maximum plasma concentration (*C*_max_)/*K*_i_ exceeded 1.0 [35,62]. After oral administration of RVT (5 g, single dose) to healthy subjects, the plasma *C*_max_ value of RVT was approximately 2.36 μM [63]. The plasma *C*_max_ value of RVT increased to 8.78 μM after a single oral dose of micronized RVT formulation (5 g) in humans [57]. The calculated values of *C*_max_/*K*_i_ were 1.48~5.49, 0.51~8.78, 0.26~0.96, and 0.24~0.90 from the inhibition constant of tRVT against CYP1A2, CYP3A, CYP2E1, and CYP2C19 (*K*_i_ = 1.6 μM, 1.0~4.6 μM, 9.1 μM, and 9.8 μM, respectively), suggesting that RVT might have FDI potential with these P450 substrate drugs at high doses. 

Here, we demonstrated the stereoselective inhibitory potential of RVT against both P450s and UGTs in HLMs. We also determined their TDI against P450s, and indirectly confirmed the formation of reactive quinone methide intermediate using glutathione as a trapping agent. tRVT stereoselectively inhibited CYP2C9, CYP3A, and UGT1A1 activities whereas cRVT had negligible inhibition on them (IC_50_ > 50 μM). In addition, tRVT time-dependently inhibited CYP1A2, CYP2C19, CYP2E1, and CYP3A with *K*_i_ shift values >2.0, while *cis*-resveratrol time-dependently inhibited CYP2C19 and CYP2E1. From the inhibition potential (IC_50_, *K*_i_, and VDI values) of RVT against P450s and UGTs, RVT at high doses might cause significant pharmacokinetic FDI with co-administered drugs predominantly metabolized by UGT1A1, CYP3A, CYP2E1, CYP2C19, and CYP1A2 in humans.

## Figures and Tables

**Figure 1 pharmaceutics-13-01419-f001:**
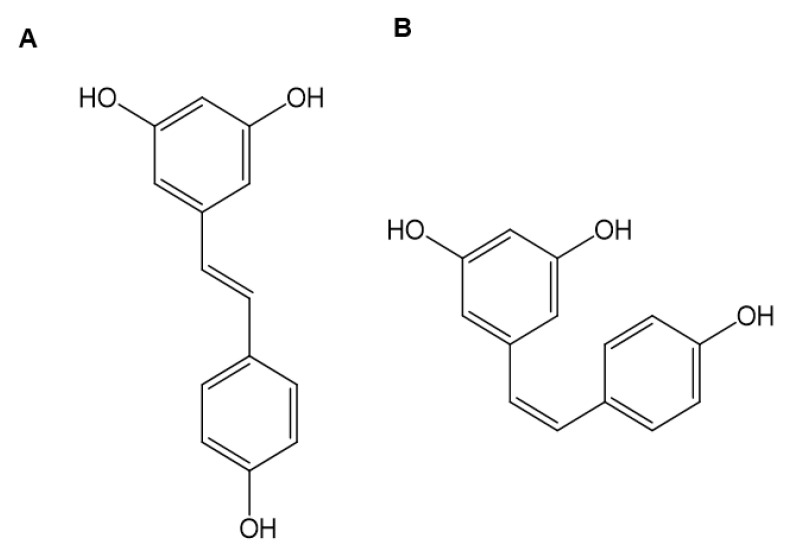
Chemical structures of *trans*-resveratrol (**A**) and *cis*-resveratrol (**B**).

**Figure 2 pharmaceutics-13-01419-f002:**
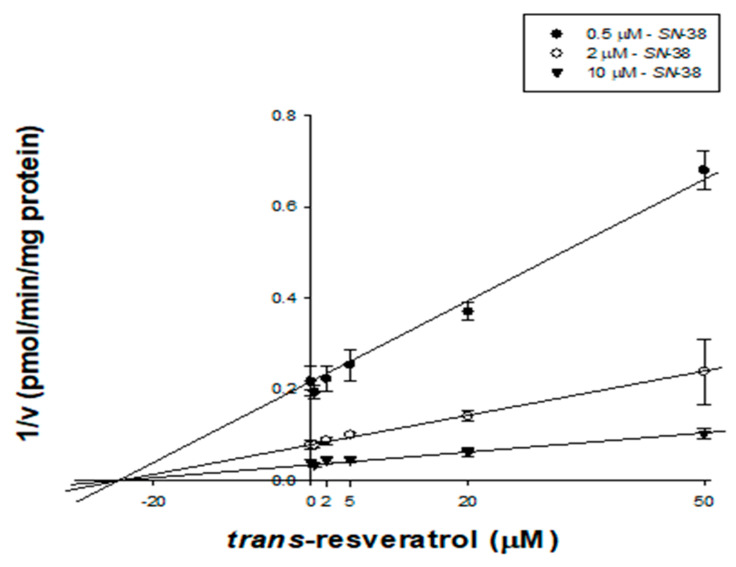
Representative Dixon plot obtained from an inhibition kinetic study of UGT1A1-mediated SN-38 glucuronidation in the presence of different concentrations of *trans*-resveratrol in pooled human liver microsomes (*n* = 3). An increasing concentration of SN-38 (0.5 (⬤), 2.0 (◯), and 10 μM (▼)) was incubated with HLMs (0.25 mg/mL) and UDPGA (1 h, 37 °C) in the presence or absence of *trans*-resveratrol (0, 0.5, 2.0, 5.0, 20, and 50 μM).

**Figure 3 pharmaceutics-13-01419-f003:**
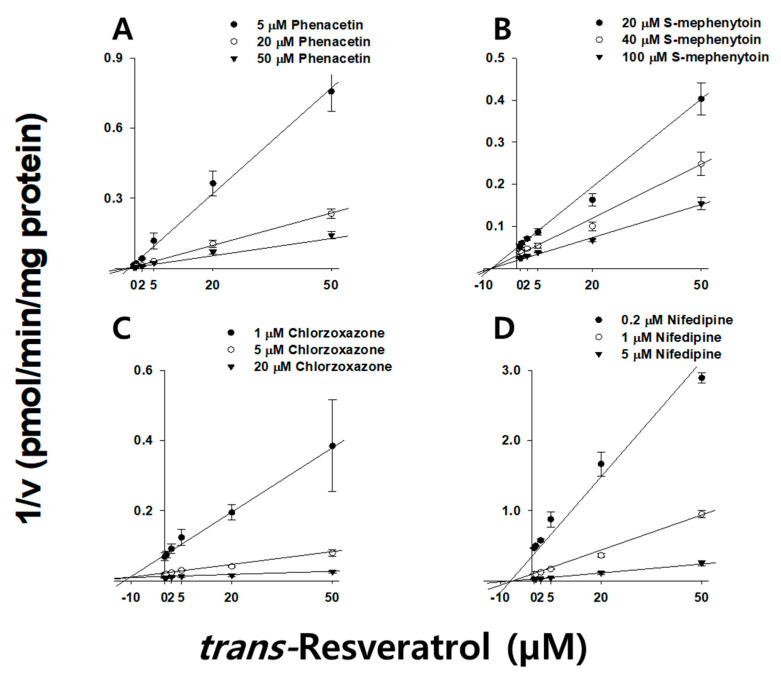
Representative Dixon plots obtained from inhibition kinetic studies of CYP1A2-mediated phenacetin *O*-deethylation (**A**); CYP2C19-mediated *S*-mephenytoin 4-hydroxylation (**B**); CYP2E1-mediated chlorzoxazone 6-hydroxylation (**C**), and CYP3A-mediated nifedipine dehydrogenation (**D**) in the presence of different concentrations of *trans*-resveratrol in pooled human liver microsomes (*n* = 3). Each *trans*-resveratrol (0, 0.5, 2.0, 5.0, 20, and 50 μM) was pre-incubated with HLMs in the presence of an NADPH-generating system (30 min, 37 °C), and then an increasing concentration of phenacetin (5 (⬤), 20 (◯), and 50 μM (▼)), *S*-mephenytoin (20 (⬤), 40 (◯), and 100 μM (▼)), chlorzoxazone (1 (⬤), 5 (◯), and 20 μM (▼)), and nifedipine (0.2 (⬤), 1 (◯), and 5 μM (▼)) was added prior to further incubation for 10 min.

**Figure 4 pharmaceutics-13-01419-f004:**
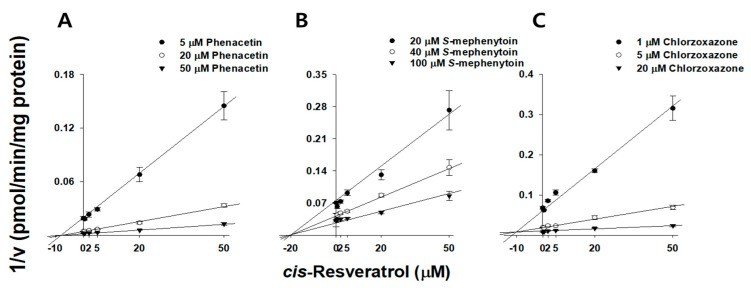
Representative Dixon plots obtained from an inhibition kinetic study of CYP1A2-mediated phenacetin *O*-deethylation (**A**); CYP2C19-mediated *S*-mephenytoin 4-hydroxylation (**B**), and CYP2E1-mediated chlorzoxazone 6-hydroxylation (**C**) in the presence of different concentrations of *cis*-resveratrol in pooled human liver microsomes (*n* = 3). Each *cis*-resveratrol (0, 0.5, 2.0, 5.0, 20, and 50 μM) was pre-incubated with HLMs in the presence of an NADPH-generating system (30 min, 37 °C), and then an increasing concentration of phenacetin (5 (⬤), 20 (◯), and 50 μM (▼)), *S*-mephenytoin (20 (⬤), 40 (◯), and 100 μM (▼)), and chlorzoxazone (1 (⬤), 5 (◯), and 20 μM (▼)) added before further incubation for 10 min.

**Figure 5 pharmaceutics-13-01419-f005:**
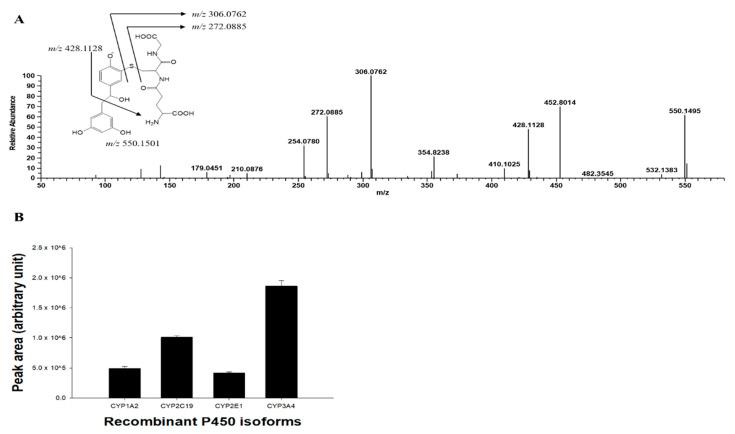
MS/MS spectrum of glutathione conjugate of resveratrol annotated with the proposed structure of fragment ions obtained by LC-HRMS analysis of the human recombinant CYP3A4 incubates of resveratrol in the presence of an NADPH- generating system and glutathione (**A**) and representative plots for the formation of the glutathione conjugate of *trans*-resveratrol by human recombinant P450 supersome (rP450) (**B**). The incubation system contained 0.1 M phosphate buffer, 1 pmol P450 supersomes, 20 μM *trans*-resveratrol, 1 mM NADPH, and 10 mM glutathione. The data are shown as mean ± S.D. (*n* = 3).

**Table 1 pharmaceutics-13-01419-t001:** Selected reaction monitoring (SRM) conditions for the major metabolites of the cytochrome P450 (P450) probe substrates and the uridine 5′-diphosphoglucuronosyl transferase (UGT) substrates used in all assays.

Enzyme	Substrate *	Concentration (µM)	Metabolite	SRM Transition (*m/z*)	Collision Energy (eV)	Polarity **
CYP1A2	Phenacetin	20	Acetaminophen	152 > 110	25	ESI^+^
CYP2A6	Coumarin	1	7-Hydroxycoumarin	163 > 107	17	ESI^+^
CYP2B6	Bupropion	3	Hydroxybupropion	256 > 238	10	ESI^+^
CYP2C8	Amodiaquine	0.1	*N-*Desethylamodiaquine	328 > 283	13	ESI^+^
CYP2C9	Diclofenac	1	4-Hydroxydiclofenac	312 > 231	15	ESI^+^
CYP2C19	*S-*Mephenytoin	40	4-Hydroxymephenytoin	235 > 150	15	ESI^+^
CYP2D6	Dextromethorphan	2	Dextrorphan	258 > 157	30	ESI^+^
CYP2E1	Chlorzoxazone	5	6-Hydroxychlorzoxazone	184 > 120	18	ESI^−^
CYP3A	Nifedipine	0.2	Dehydronifedipine	345 > 284	30	ESI^+^
	Midazolam	0.1	1′-Hydroxymidazolam	342 > 203	28	ESI^+^
IS	Trimipramine	0.07	-	295 > 100	17	ESI^+^
UGT1A1	SN-38	0.5	SN-38 glucuronide	569 > 393	30	ESI^+^
UGT1A3	CDCA	2	CDCA-24 glucuronide	567 > 391	20	ESI^−^
UGT1A4	TFP	0.5	TFP-β-d-glucuronide	584 > 408	30	ESI^+^
UGT1A6	*N-*ASER	1	*N-*ASER-β-d-glucuronide	395 > 219	10	ESI^+^
UGT1A9	MPA	0.2	MPA-β-d-glucuronide	495 > 319	25	ESI^−^
UGT2B7	Naloxone	0.2	Naloxone-β-d-glucuronide	504 > 310	30	ESI^+^
IS	Estrone-β-d-glucuronide	0.25	-	445 > 269	35	ESI^−^

*** SN-38: 7-Ethyl-10-hydroxycamptothecin; CDCA: Chenodeoxycholic acid; TFP: Trifluoperazine; *N*-ASER: *N-*Acetylserotonin, and MPA: Mycophenolic acid. ** ESI: Electrospray ionization (ESI) interface to generate protonated ion [M+H]^+^ or deprotonated ion [M-H]^−^.

**Table 2 pharmaceutics-13-01419-t002:** Inhibitory effects (IC_50_ values) of *trans*- and *cis*-resveratrol against nine cytochrome P450 isoforms (P450s) in HLMs (*n* = 3).

P450 Enzyme	Substrate	*Trans-*resveratrol (µM)			*Cis-*resveratrol (µM)		
		Without Preincubation	With Preincubation (30 min)	IC_50_ Shift	Without Preincubation	With Preincubation (30 min)	IC_50_ Shift
1A2	Phenacetin	12.9 ± 2.50	1.21 ± 0.20	10.80	14.0 ± 4.10	7.20 ± 0.50	1.90
2A6	Coumarin	>50.0	>50.0	–	>50.0	48.3 ± 17.0	–
2B6	Bupropion	34.8 ± 7.20	40.1 ± 14.9	0.90	28.1 ± 13.9	32.8 ± 6.10	0.90
2C8	Amodiaquine	>50.0	35.2 ± 6.40	>1.40	>50.0	>50.0	–
2C9	Diclofenac	36.2 ± 3.30	23.8 ± 4.10	1.50	>50.0	>50.0	–
2C19	*S-*Mephenytoin	48.4 ± 10.9	8.13 ± 1.61	6.00	46.5 ± 16.1	19.8 ± 4.20	2.30
2D6	Dextromethor-phan	>50.0	32.8 ± 3.20	>1.50	>50.0	>50.0	-
2E1	Chlorzoxazone	>50.0	9.82 ± 2.21	>5.00	>50.0	10.8 ± 2.90	>4.60
3A	Midazolam	29.2 ± 4.50	16.6 ± 4.00	1.80	>50.0	>50.0	-
	Nifedipine	19.6 ± 2.30	5.62 ± 0.60	3.50	>50.0	>50.0	-

**Table 3 pharmaceutics-13-01419-t003:** Inhibitory effects (IC_50_ values) of *trans*- and *cis*-resveratrol against six uridine 5′-diphospho-glucuronosyltransferase isoforms (UGTs) in HLMs (*n* = 3).

UGT Enzyme	Substrate *	*Trans-*Resveratrol	*Cis-*Resveratrol
1A1	SN-38	9.57 ± 1.60	>50.0
1A3	CDCA	>50.0	>50.0
1A4	TFP	>50.0	>50.0
1A6	*N-*ASER	>50.0	>50.0
1A9	MPA	>50.0	>50.0
2B7	Naloxone	>50.0	>50.0

*** SN-38: 7-Ethyl-10-hydroxycamptothecin; CDCA: Chenodeoxycholic acid; TFP: Trifluoperazine; *N*-ASER: *N*-Acetylserotonin, and MPA: Mycophenolic acid.

**Table 4 pharmaceutics-13-01419-t004:** *K*_i_ values for the inhibition of CYP1A2-mediated phenacetin *O*-deethylatoin, CYP2C19-mediated *S*-mephenytoin 4-hydroxylation, CYP2E1-mediated chlorzoxazone 6-hydroxylation, CYP3A-mediated nifedipine dehydrogenation, CYP3A-mediated midazolam hydroxylation, and UGT1A1-mediated SN-38 glucuronidation activities in human liver microsomes by *trans*- and *cis*-resveratrol.

Enzyme	Substrate	*trans-*Resveratrol				*cis-*Resveratrol			
		Without Preincubation		With Preincubation (30 min)		Without Preincubation		With Preincubation (30 min)	
		*K*_i_ (μM)	Mode of Inhibition	*K*_i_ (μM)	Mode of Inhibition	*K*_i_ (μM)	Mode of Inhibition	*K*_i_ (μM)	Mode of Inhibition
CYP1A2	Phenacetin	13.8 ± 1.70	Competitive	1.62 ± 0.11	Noncompetitive	9.19 ± 1.41	Competitive	9.46 ± 0.63	Noncompetitive
CYP2C19	*S-*Mephenytoin	-	-	9.78 ± 0.60	Noncompetitive	-	-	21.8 ± 6.10	Noncompetitive
CYP2E1	Chlorzoxazone	-	-	9.11 ± 1.62	Competitive	-	-	8.08 ± 1.43	Competitive
CYP3A	Nifedipine	23.6 ± 1.90	Noncompetitive	4.60 ± 0.11	Noncompetitive	-	-	-	-
	Midazolam	23.8 ± 2.30	Noncompetitive	1.02 ± 0.22	Uncompetitive	-	-	-	-
UGT1A1	SN-38 *	27.4 ± 3.60	Noncompetitive	-		-		-	

*** SN-38: 7-Ethyl-10-hydroxycamptothecin.

## Data Availability

All data in this study have been included in this manuscript.

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
