# Peer review of "Comprehensive Investigation of Stereoselective Food Drug Interaction Potential of Resveratrol on Nine P450 and Six UGT Isoforms in Human Liver Microsomes"

_pharmaceutics, 2021, doi:10.3390/pharmaceutics13091419_

Round 1
Reviewer 1 Report
Seung-Bae Ji et al investigated the stereoselective food-drug interaction potential of resveratrol on 9 CYP and 6 UGT isoforms in human liver microsomes.
- The authors need to correct the spelling mistakes in the manuscript. Line 67, 212, table 1.
- Why the authors performed the inhibition studies using a cocktail substrate approach instead of an individual substrate approach? The individual substrate approach is more reliable and generates accurate results compared to the cocktail approach. The authors should mention the information in the manuscript.
- The authors repeated sentences at several places (lines 155, 164, 190, 215) in the methods section. These sentences can be mentioned in one place to avoid repeats in the method section.
- Table 1 should be included in the LC-MS/MS section (2.7).
- In figure 3, why the authors didn’t show the Dixon plots for resveratrol using midazolam? The authors should include the Dixon plots for resveratrol using midazolam in figure 3.
- In figure 5b, the x-axis legend should be mentioned in detail.
Reviewer 2 Report
Review report
Manuscript ID: 1337657
Title: Comprehensive Investigation of Stereoselective Food Drug Interaction Potential of
Resveratrol on Nine P450 and Six UGT Isoforms in Human Liver Microsomes
Brief summary:
The manuscript aims at evaluating the stereoselectivity of the food-drug inhibitory potential of
resveratrol on nine cytochrome P450s and six uridine 5’-diphosphoglucuronosyl transferases.
The main innovation of the paper relies on the description of cis-resveratrol inhibitory mechanisms
and kinetics using human liver microsomes, to ultimately predict the potential of food-drug
interactions.
Broad comments:
The manuscript meets the scope of the journal as it intends to evaluate the potential of food-drug
interactions of resveratrol using in vitro methodologies. The manuscript is well-structured, and the
obtained data support the main conclusions. The manuscript is easy to follow due to the simple
and straightforward way the results are presented and discussed. In a positive note to improve
the manuscript, I suggest to the authors to focus the abstract on the innovative data obtained with
cis-resveratrol and to improve the quality of tables and figures. Regarding figures, I also believe
that Figure 2 and 4 may be moved to supplementary material to facilitate the task for the readers.
Specific comments:
Please find below some minor points to improve the manuscript:
a. Line 47/48: Please include a reference in the sentence: “In 2019, global dietary
supplements market was 47 valued at $ 163 billion.”
b. Table 1: The clarity of the table could be improved if formatted considering the total width
of the page.
2
c. Line 185 and 188: Replace 37ºC by 37 ºC and 4ºC by 4 ºC, respectively. Please check
the missing space between number and degree sign throughout the manuscript.
d. Line 211 and 225: The reference 58 and 60 is not well cited according to the journal
indications. Please check.
e. Tables and Figures: Include italic style in trans- and cis- prefix.
f. Section 3.3: Replace r2 by r2
.
g. Line 377: I believe the authors indicate (Figure 2) instead of (Table 4) by mistake.
h. Line 400: It is not clear why cRVT is unlike tRVT as both display TDI inhibition of CYP2C19
and CYP2E1. Please clarify.
